# Recent Development of Drug Delivery Systems through Microfluidics: From Synthesis to Evaluation

**DOI:** 10.3390/pharmaceutics14020434

**Published:** 2022-02-17

**Authors:** Zhiyuan Ma, Baicheng Li, Jie Peng, Dan Gao

**Affiliations:** State Key Laboratory of Chemical Oncogenomics, Guangdong Provincial Key Laboratory of Chemical Biology, Tsinghua Shenzhen International Graduate School, Shenzhen 518055, China; ma-zy20@mails.tsinghua.edu.cn (Z.M.); lbc20@mails.tsinghua.edu.cn (B.L.); pengj21@mails.tsinghua.edu.cn (J.P.)

**Keywords:** microfluidic, drug delivery system, carrier-free, in vitro model, micro-reservoir, microneedles

## Abstract

Conventional drug administration usually faces the problems of degradation and rapid excretion when crossing many biological barriers, leading to only a small amount of drugs arriving at pathological sites. Therapeutic drugs delivered by drug delivery systems to the target sites in a controlled manner greatly enhance drug efficacy, bioavailability, and pharmacokinetics with minimal side effects. Due to the distinct advantages of microfluidic techniques, microfluidic setups provide a powerful tool for controlled synthesis of drug delivery systems, precisely controlled drug release, and real-time observation of drug delivery to the desired location at the desired rate. In this review, we present an overview of recent advances in the preparation of nano drug delivery systems and carrier-free drug delivery microfluidic systems, as well as the construction of in vitro models on-a-chip for drug efficiency evaluation of drug delivery systems. We firstly introduce the synthesis of nano drug delivery systems, including liposomes, polymers, and inorganic compounds, followed by detailed descriptions of the carrier-free drug delivery system, including micro-reservoir and microneedle drug delivery systems. Finally, we discuss in vitro models developed on microfluidic devices for the evaluation of drug delivery systems, such as the blood–brain barrier model, vascular model, small intestine model, and so on. The opportunities and challenges of the applications of microfluidic platforms in drug delivery systems, as well as their clinical applications, are also discussed.

## 1. Introduction

Many therapeutic drugs might face problems such as low biodistribution, limited solubility, poor absorption, and drug aggregation. Drug delivery systems (DDSs) aiming to transport therapeutic drugs to desired areas, including tissues, organs, cells, subcellular organs, and so on, are a usual approach to improving pharmacological activity [1]. Recently, various drug carriers have been developed for transporting drugs to the target site and protecting them from improper degradation, which ensures maximum drug efficacy and reduces side effects [2]. However, the development of DDSs is an arduous, multi-step process involving mass production, chemical characterization, toxicity testing, and clinical trials, which prevents rapid application from the lab to the clinic. Furthermore, the existence of internal barriers in our body, such as the mucosal diffusion barrier and the cellular permeability barrier, may preclude the efficiency of conventional drug delivery routes. Conventional drug delivery methods resort to different routes of administration such as hypodermic injections, oral administration, and inhalation to enter our body, but they still show a weak ability to help some pharmaceutical molecules traverse barriers and control their release to the desired location [3]. Fortunately, emerging nanotechnologies hold significant promise in developing the next generation of DDSs, aiming to propose more powerful and self-regulated delivery tools [3].

Known as a “lab on a chip” (LOC), microfluidics technology represents a rapidly growing versatile technology for manipulating nanoliter amounts of fluids within microscale channels. It has emerged in recent years as a distinct new area of research, especially in chemistry, medicine, and physical sciences. With its unprecedented advantage in the precise control of fluid, it not only can improve the fabrication efficiency and quality of drug delivery, but also shows great potential for the predictive power of preclinical drug carrier testing through biomimetic microfluidic platforms [4]. Compared to the conventional methods, the microfluidics platform enables continuous, scalable, and reproducible production and improves the uniformity, yield, and batch-to-batch reproducibility of nanoparticles due to its miniaturization and automation [5]. To date, many microfluidics platforms [6,7] have been proposed for integrating multiple processes and functions, varying from synthesis to testing. Studies on microneedle-based systems also show great potential for future applications [8,9]. Although the fantastic manipulating capability enables the mass production of emulsion templates for synthesizing microparticles, the issues of industrial-scale production using microfluidics platforms are still under consideration. Several bouts of research have shown great effort put towards sizing up the volume to milliliter scales and further improving the throughput by combining multiple reactions in parallel [10,11,12].

Owing to the miniaturization of the fluidic environment, microfluidics has brought a highly controllable, reproducible, and scalable fabrication platform to the production of drug carriers. Among the presently developed drug carriers, nanoparticles have shown excellent properties of improving the therapeutic index, reducing side effects, and enhancing uptake and penetration [13,14]. The development of microfabrication technologies has offered capabilities to produce nanoparticles in a controllable and reproducible manner, including flow focusing, template assembly, and droplet technology. The physicochemical properties such as size, shape, and composition of nanoparticles can be precisely controlled at a larger dynamic range [15], allowing for an increase in drug transport efficacy, release profile, and elimination during treatment. Furthermore, high-throughput fabrication is realizable by parallelization control and reproducible scale-up production. Apart from the fabrication of nanocarriers for drugs, carrier-free DDSs have also been widely used as a direct drug delivery method. By storing the drugs directly in the microfluidic chip, it can further improve efficiency and combat the problems that carrier-based delivery systems have not covered [16]. For example, carrier-free DDSs do not need any redundant synthesis steps, and they can ensure that drugs release precisely into target sites [17]. Without changing the dose or shape of the drugs, the system can not only achieve the zero-order release of the drug, but also lead to a controllable and reproducible release profile [2]. Furthermore, carrier-free DDSs can minimize side effects. 

In addition to the advantages with respect to the establishment of controlled DDSs, the great potential of microfluidics for mimicry of the complex biological environment in vitro provides a powerful tool for the validation and further assessment of the biocompatibility and efficacy of drug delivery in a biological context. Several recent reviews have summarized microfluidic-based cell culture models to engineer micro-sized human tissues and organs. These models can help accelerate drug development by resolving the discrepancies in animal models in certain aspects. Besides, they can bring great convenience in terms of the observation of the drug delivery process through real-time imaging and in vitro microscopic observation technology. 

In this review, we first highlight the application of microfluidics technology in the establishment of a variety of emerging DDSs over the past five years, including the fabrication of nanoparticle-based systems and carrier-free systems, and then summarize biomimetic in vitro models on microfluidic chips to assess DDSs. Overall, recent innovations and advances in microfluidic-based systems are expected to accelerate the transition of new DDSs to clinical evaluation.

## 2. Synthesis of Drug Delivery Carriers on Microfluid Chips

Due to the excellent ability to manipulate nanoliter flows, microfluidics has been extensively applied in the fabrication of nanoparticles as well as the preparation of nanoparticle-based DDSs. Generally, the nanocarriers include lipid-based nanoparticles, polymeric nanoparticles, and inorganic nanoparticles. A big challenge for conventional nanoparticle systems is that it is hard to control the homogeneity of the particle size and shape. Furthermore, the safety and biocompatibility of the nanomaterials themselves are extremely important factors that need to be focused on [14]. Recent research in microfluidic technologies has shown great potential in nanoparticles’ synthesis, characterization, and assessment for drug delivery application [15]. They can provide a controllable and reproducible means for nanoparticle production, and the sizes, shapes, and surface compositions can be precisely adjusted to meet the needs of different kinds of drugs. Several platforms have been put forward for nanoparticle synthesis, which can be divided into single-phase systems and multiphase systems [18]. In single-phase systems, a continuous laminar flow of fluid provides a homogenous environment for diffusion and fast mixing and stirring, ensuring the nanoparticles’ nucleation and growth. As for the multiphase system, the discrete segments caused by immiscible fluids will act as individual reaction chambers where mixing and synthesizing are generated. In this part, we will discuss the synthesis of nanoparticle-based DDSs on microfluidic devices. 

### 2.1. Lipid-Based Nanoparticles

Having a similar structure to the cell membrane, lipid-based nanoparticles exhibit numerous merits in terms of biocompatibility, penetration ability, ease of surface modification, and high drug-loading capacity, which have been popularly applied in DDSs [14]. The size of lipid-based nanoparticles is a significant factor for drug delivery efficiency and therapeutic efficiency. However, conventional studies suffer from the complicated processes of preparing the liposomes, because a post-processing step is required to maintain uniformity for the liposomes. Microfluidic platforms have gained substantial attention for overcoming these disadvantages. Studies on the fabrication of liposomes in microfluidics date back to 2004 [19], using flow-focusing methods to assemble the liposomes automatically through precise fluid control. Generally, the size of liposomes synthesized in microfluidics is controlled by the flow rate of the solutions, the mixing efficiency, and the flow rate ratio. To enhance nanoparticles’ uniformity, Le et al. [20] proposed an acoustically enhanced micromixer, which has been applied in the synthesis of highly uniform nanoscale budesonide without the addition of stabilizers. A 3D microfluidic geometry structure has been employed for rapidly and efficiently mixing the solvent and antisolvent phases, and the results are quite inspiring, with a mean diameter of less than 150 nm, reducing the size by 40-fold in comparison to the earlier works. Furthermore, a more precise size control approach [21] capable of fulfilling 10 nm intervals ranging from 20 to 100 nm (Figure 1E) was also proposed. The basic structure of this approach is 20 sets of the baffle mixer structure, which is similar to a zigzag-shaped microchannel and can control the size of the lipid nanoparticle as well as improve the drug delivery efficiency. However, limitations from the small dimensional scale of microfluidic reactors, such as low production rates, short equipment life, and high manufacturing costs, hindered their further industrial application. To overcome these difficulties, reactions with larger dimensions were needed. For example, the Yanar research group [22] developed millimeter-scale flow reactors capable of synthesizing liposomes at the industrial scale with high production rates and high stability. Apart from flow-focusing methods, nanoscale liposomes with a higher range of diameters can be generated in a micromixer structure [16]. At present, the micromixer structures used to improve mixing efficiency include Y-type, T-type, and staggered herringbone micromixers [23]. Specifically, SHM can provide the most efficient mixing and has been widely used for nanoparticle production. As the synthesis efficiency and the size of nanoparticles would be affected by the mixer dimensions, the exploration of the optimal design of passive micromixers is critical. Wang et al. [24] built a library of thousands of different randomly designed mixers and used the non-dominated sorting genetic algorithm II (NSGA-II) to optimize the random chips in order to achieve Pareto efficiency (Figure 1D). In addition, the lipid concentration, composition [25], and solvent will affect the liposomes’ formation. With the adjustment of flow rate ratio (FRR), the compositions of nanoparticles are changeable. Lin et al. [26] investigated the effects of FRR on liposomes’ size and drug-loading efficiency. They found that the loading efficiency of hydrophilic drugs had a positive linear correlation with the FRR, and the maximum drug-loading efficiency could reach up to 90%. Balbino et al. [27] designed an integrated microfluidic device with two different regions for efficient generation of liposomes and lipoplexes in a continuous flow, which significantly reduced the steps of lipoplex synthesis (Figure 1A). In vitro transfection assays showed that microfluidic-obtained lipoplexes had the same transfection ability as the conventionally obtained lipoplexes.

### 2.2. Polymeric Nanoparticles

Polymeric nanoparticles, including natural polymers, synthetic polymers, and so on, have been widely used in the field of drug delivery. The main mechanism is that polymeric nanoparticles can interact with mucus through electrostatic, van der Waals, hydrophobic, or hydrogen-bonding interactions, resulting in a long residence time for drug absorption. Among various polymeric nanoparticles, poly lactate glycolic acid (PLGA) is the most commonly used polymeric material. PLGA is a family of linear copolymers that are non-toxic and biodegradable and which have been widely used for medical applications. The surface of PLGA can be easily modified with different kinds of active groups to realize a targeting function towards specific disease [28]. With the advantage of improved control over size, size distribution, and morphologies, a microfluidic-assisted nanoprecipitation strategy was widely employed. Leung et al. [29] presented a hydrodynamic flow-focusing-based device for nanoprecipitation, and the surfactant-free curcumin-encapsulated PLGA nanoparticles were successfully synthesized with diameters ranging from 30 to 70 nm [29]. Apart from the diffusion methods such as hydrodynamic flow focusing mentioned above, droplet-based synthetic methods were also a powerful choice for polymeric nanoparticle synthesis. Generating reversed microemulsion droplets and controlling the evaporation of organic solvent are the key steps. Yu et al. [30] have established a one-step synthesis of monodisperse functional polymeric microspheres on a droplet-based microfluidic device. This method relies on the formation of a stable microemulsion first and then a rapid evaporation of the oil from the droplets during the synthesis process for the formation of polymeric microspheres and the inclusion of functional materials. Pavithra et al. [31] further studied the factors that may impact the evolution from droplets into particles. They proposed a particle generation technique by making the droplet containing polymeric solution dissolve into the surrounding partially immiscible liquid to drive the particles’ formation (Figure 1B). A digital microfluidics platform was also used for the manipulation of nanoparticles. Differing from the conventional droplet-based methods, the digital microfluidics platform is on an open surface and enables individual control of each droplet, offering great power in parallel processing [32]. Alsaeed et al. [33] proposed a digital microfluidics platform for PLGA synthesis and characterization. The diameters of the monodisperse PLGA nanoparticles were uniform at the small size of 115 nm.

### 2.3. Inorganic Nanoparticles

With their unique physicochemical properties, such as surface plasmon resonance, inorganic particles including quantum dots, graphene, iron oxide, and silica nanoparticles show superior performance in drug delivery. Controlled synthesis and surface engineering of inorganic nanomaterials enable a high flexibility for functional design. As one of the widely used nanomaterials for biological imaging, quantum dots also play an important role in targeted therapy and drug delivery. Li et al. [34] reported quantum dots functionalized with multiple paired α-carboxyl and amino groups structurally mimicking large amino acids, which can bind to large neutral amino acid transporter 1 (LAT1). LAT1 is selectively and highly expressed in tumor cells, so it serve as a promising drug carrier with good selectivity for tumor imaging. Furthermore, a triple conjugated system by conjugating transferrin and loading two anticancer drugs onto carbon dots with sizes of 3.5 nm has been developed to solve the difficulty of entering malignant brain tumors [35]. Results showed that this system could deliver the loaded chemotherapeutic drugs into glioblastoma brain tumor cells via receptor-mediated endocytosis. However, the parameters and conditions for the fabrication of quantum dots and the quality of the synthesis reaction should be precisely controlled. The generation of quantum dots through microfluidic technology can obtain a controllable surface area. Specifically, with precise control over the parameters (e.g., injection speed and synthesis size), continuous-flow microfluidics have attracted widespread interest. Guidelli et al. [36] developed a microfluidic reactor for the synthesis of ZnSe quantum dots (Figure 1C). Through precisely controlling the flow rate and reaction times, ZnSe nanocrystal nucleation and growth could be made to happen in two different stages. Furthermore, Baek et al. [37] presented a multistage microfluidic platform to achieve the continuous synthesis of complex nanostructures. This platform consisted of a series of custom-designed chip reactors, which can be easily reconfigured by reconnecting different chip reactors. Therefore, this platform is capable of precise control of heating profiles and flow distribution while conducting multistep reactions. Apart from quantum dots, mesoporous silica nanoparticles (MSNs) are newcomers in this field. Mesoporous silica materials were first reported in the 1990s and, having the presence of silanol groups at their surface, showed good bioactive behavior, and their physical-chemical properties make them a good choice for drug delivery. A multitargeted delivery system shown in Figure 1F has been developed for sequential cell–organelle targeting with two different therapeutic cargoes [38]. The secondary targeting units and the drug were loaded onto the holes of the MSNs, and the primary targeting unit was modified on the surface of the MSNs. This example demonstrated great potential for forming a multifunctional DDS. Yan et al. [39] proposed an MSN fabrication platform using a pH/redox-triggered method. The precise control of fluid ensured the control of the loading degree, solubility, and stability of the hydrophobic drug. Microfluidics also ensure the controllable synthesis of nanomaterials with desired shapes for various applications. Hao et al. showed an example of the generation of functional MSNs with tunable aspect ratios. A miniaturized spiral-shaped microchannel with one outlet and two inlets was designed for the synthesis of MSNs. By changing the flow rate and the concentrations of the reactants, MSNs’ aspect ratios and diameters can be accurately tuned. Other studies [40,41] supported that the porous structures can be easily and precisely adjusted by manipulating the flow of solutions during the microfluidic emulsification. There is no doubt that the great advantages of microfluidics endow the MSNs with more advanced physicochemical and biological properties. Their applications in drug loading and targeted delivery have shown encouraging results.

**Figure 1 pharmaceutics-14-00434-f001:**
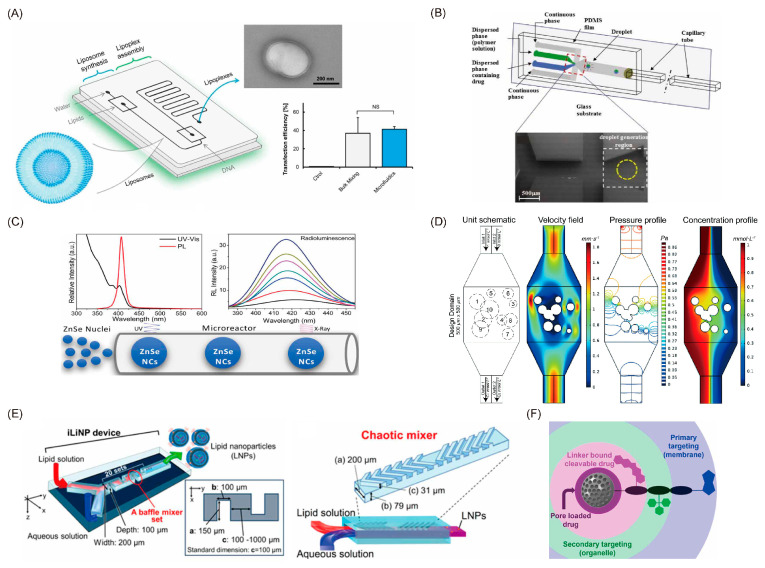
Different approaches for the fabrication of nanoparticles. (**A**) Schematic of the microfluidic device for one-step formation of plasmid DNA (pDNA)/cationic liposome (CL) lipoplexes and its in vitro efficacy. (**B**) Schematic of the flow-focusing device used to engineer and monitor Janus drug particle formation. (**C**) Schematic of the microfluidic reactor used for the synthesis of ZnSe quantum dots. (**D**) The simulated microfluidic mixer unit with two inlets and two outlets. (**E**) Three-dimensional views and top views of the iLiNP device. (**F**) The strategy employed for dual-targeted two-drug nanocarriers based on mesoporous silica nanoparticles. (**A**) Balbino et al. [27]; (**B**) Sundararajan et al. [31]; (**C**) Guidelli et al. [36]; (**D**) Wang et al. [24]; (**E**) Kimura et al. [21]; (**F**) Castillo et al. [38].

## 3. Microfluidic Chip for Carrier-Free Drug Delivery

Microfluidic chips can not only be used to synthesize a variety of complicated nanodrug carriers, but can also characterize the efficiency of drug delivery directly through various in vivo models [42,43,44]. For nano drug carrier systems, a large amount of expensive drugs may be used, and the synthesis of drug carriers may face various challenges. For instance, the composition of carriers must be strictly controlled to ensure the efficacy of drugs. For microfluidic carrier-free DDSs, [2,45] direct drug delivery can ensure drug release to the special application site quickly and effectively. Moreover, many of the carrier-free drug delivery systems can avoid immune rejection and reduce the toxicity caused by non-drug carriers [2,46,47]. Carrier-free direct drug delivery can be roughly divided into two categories: drug delivery based on microneedles (MNs) and drug delivery based on micro-reservoirs [17,48,49,50].

### 3.1. MNs-Based System

In recent decades, research has explored a novel transdermal drug delivery strategy, MNs, which can overcome some limitations of traditional oral and subcutaneous administration. MNs can easily pass through the stratum corneum (SC) of the skin, a barrier to drug molecules. Besides, patients can easily administer drugs by themselves through MNs. Compared with subcutaneous administration, the length of MNs can be controlled to be between 10 μm and 1 mm, which is only a small fraction of the subcutaneous administration needle, so that they can achieve painless treatment due to their short size [51,52,53,54,55,56]. Compared with traditional oral administration, MNs’ percutaneous administration can avoid the poor absorption caused by the degradation of drugs through the metabolic system [52,57,58]. So far, there are several kinds of MNs reported for drug delivery application, such as solid MNs, coated MNs, dissolving MNs, hollow MNs, and hydrogel-based MNs [2,59,60].

#### 3.1.1. Solid MNs

Solid MNs are made of different materials such as silicone, metals, and polymers, which are manufactured by 3D printing, digital light processing (DLP), stereolithography, laser cutting, and chemical or mechanical technologies into specific shapes and lengths. The working principle of solid MNs involves a two-step process. The solid MNs are firstly used to create a micro-channel on the surface of the skin, and then the drugs are coated onto the skin surface to make the drug diffuse passively to achieve the therapeutic effect [59,60]. The effect of DDSs of solid MNs depends on the shape and length of the MNs, and many efforts have been made to improve the drug delivery efficiency of the solid MNs. For example, Narayanan et al. [61] developed solid silicon MNs and explored their best aspect ratio, mechanical strength, and fracture strength conditions by using single-step lithography and an anisotropic wet etching tetramethylammonium hydroxide (TMAH) process to achieve the best drug delivery effect. They successfully synthesized solid MNs with an average height of 158 μm, width of 110.5 μm, aspect ratio of 1.43, tip angle of 19.4°, and tip diameter of 0.40 μm. Additionally, the microhardness value is 44.4 (HRC), which is 52.2 times higher than the skin’s ultimate tensile strength (UTS), resulting in a great improvement of its percutaneous drug administration ability. Similarly, Li et al. [62] built solid MNs using polylactic acid (PLA) materials and systematically explored the mechanical strength and biodegradable ability. They found that the MN size, drug concentration, viscosity of the drug solution, and drug administration time on the skin have great influence on administration efficiency. Under the optimized conditions, insulin could be highly efficiently delivered to the skin by the MNs.

#### 3.1.2. Coated MNs

The coated MNs are coated with drugs and biodegradable materials directly on the surface of the MNs. When the MNs enter the skin of a patient, the drugs will be dissolved and released into the body to achieve the purpose of treatment. Storing drugs by coating them onto the surface of MNs can enhance their long-term stability [59,60]. In recent years, a large quantity of research focused on the realization of combined drug delivery or drug combination adjuvant administration. For example, DeMuth et al. [63] used layer-by-layer coating techniques to realize multi-loading and multi-functional effects of DNA vaccines for the first time. Both DNA and its adjuvants were accumulated in this MN. HaeYong et al. [64] developed an intradermal pH1N1 DNA vaccine delivery platform using MNs coated with polymers containing polylactic acid co glycolic acid/polyethyleneimine (PLGA/PEI) nanoparticles (NPs) (Figure 2A). The performance of MNs coated with polymer particles via intramuscular polymer delivery and the naked pDNA vaccine in porcine skin were compared. The results indicated that the polyplex-coated MNs had a better performance towards immune response. Furthermore, to ensure the accurate control of drug dosage, the drugs coated onto MNs need to be strictly controlled. Andreas et al. [65] proposed a coated-MN-based transdermal DDS through the usage of evaporation-induced droplet transport. As shown in Figure 2B, a rough amount of the liquid containing drugs was placed on a perforated metal plate, and when the evaporated liquid reached a certain small volume, it penetrated the MN surface along the hole. Then, the MN patch stuck into the skin and left the drug behind after removing the needle. The system could ensure the accurate dosage of the drug by controlling the time of the needle exposure to the solution; the surface tension matching the degree between the needle material and the liquid; the characteristics of the liquid material; and the needle structure.

#### 3.1.3. Dissolving MNs

Dissolving MNs are composed of biodegradable and biocompatible materials such as polyvinyl alcohol (PVA) and polyvinylpyrrolidone (PVP). Drug release is mainly controlled by dissolving the materials when the MNs enter the skin [59,60]. Compared with other types of MNs, the dissolving MNs can be dissolved in the body, which means there is no risk of leaving hard residues behind. However, this kind of MN takes minutes to dissolve, and its mechanical strength is relatively weak, resulting in insufficient penetration into the skin. In order to prepare a dissolving microneedle with sufficient mechanical strength and good degradation ability, Cha et al. [66] built a polylactic acid (PLA) dissolving MN. Two different penetration and dyeing methods on pig skin successfully simulated the effect of transdermal DDSs. Guo et al. [67] also reported a novel nanostructured lipid carriers (NLCs)-loaded dissolving MN to achieve an efficient drug delivery. As shown in Figure 2C, they used NLCs as an aconitine (ACO) barrier and then embedded it in polyvinylpyrrolidone-based dissolving MNs by an ultraviolet cross-linking method. In vivo micro-dialysis proved that the MN device had an obvious inhibitory effect on paw swelling and inflammation in adjuvant-induced arthritis model rats and resulted in the improvement of the ACO-induced arrhythmia. To accelerate the dissolution of the dissolving MNs, Zhao et al. [68] built a sodium hyaluronate (HA) fast-dissolving MN by two casting methods to improve the photodynamic therapy (PDT) efficacy of a subcutaneous tumor (Figure 2D). Each microneedle can load 122 μg 5-aminolevulinic acid (ALA) in the tip; the drug loading amount was greatly improved and waste of drugs was avoided. When the needle is inserted into the skin, the MN is dissolved and the drug released. Compared to the 66% tumor inhibition rate of traditional ALA injection, the developed MNs could increase the tumor inhibition rate to 97%, and about 75% of the MN was dissolved within only 4 min.

#### 3.1.4. Hydrogel MNs

The hydrogel MNs are constructed with swelling polymer materials. This MN can be swelled in the body, and then, the expansion gap will form a 3D network to let the stored drug release. The efficiency of hydrogel MN drug delivery depends on the type of materials prepared, the mechanical strength, and the crosslinking density of the swollen hydrogel network [59,60]. The materials for the fabrication of hydrogel MNs are the most important factor for drug delivery efficiency. Recently, a lot of novel materials have been explored for the fabrication of MNs. For example, John G. et al. [69] first proposed a photo-response stimulus hydrogel MN array system for the delivery of ibuprofen. The MN arrays were prepared from 2-hydroxyethyl methacrylate (HEMA) and ethylene glycol dimethacrylate (EGDMA) by micro-molding, with good mechanical properties. The system could load up to 5% (*w*/*w*) ibuprofen in the photo-responsive 3,5-dimethoxybenzoin compounds (Figure 2E). Similarly, Chen et al. [70] developed a novel enzyme-free polymeric component MN array patch, which consisted of a boronate-containing hydrogel semi-interpenetrating network with biocompatible silk fibroin. The semi-interpenetrating network gel is composed of two components, 4-(2-acrylamidoethylcarbamoyl)-3-fluorophenylboronic acid (AmECFPBA, pKa 7.2) and an acrylamide derivative, which could enhance the glucose response to insulin. Crystalline silk fibroin [71] extracted from silkworms was used as matrix stiffener to enhance the skin penetration ability of the MNs.

#### 3.1.5. Hollow MNs

The hollow MNs are made up of several different materials such as glass, silicone, and metals. Hollow MNs are the same as those used in traditional subcutaneous administration. When the MNs enter the skin, the drugs are released into the human body through the hollow pores of the MNs. Compared with other types of MNs, they can accurately deliver high-dose drugs without changing their formula [59,60], but they face difficulties due to their insufficient mechanical strength [72]. Hollow MNs can be built by various methods, such as 3D printing [71], micro-molding, dipping, and so on [53]. For example, Anika et al. [73] fabricated a hollow MN system directly inside bulk PMMA chips for two-photon polymerization applications by femtosecond laser technology. Their single-laser system provided a more efficient and easy process to build an MN system than traditional methods. Besides, the hollow MNs can be combined with a variety of applicators to achieve the purpose of drug delivery, such as photothermal release, magnetic pulse, and so on [53]. Jayaneththi et al. [74] introduced a magnetic polymer composite (MPC) MN transdermal DDS with a single stainless steel hollow microneedle (Figure 2F). When the external magnetic pulse was applied, the MPC diaphragm moved the air into the auxiliary chamber. After the magnetic pulse disappeared, the pressure in the auxiliary chamber rose, and the high pressure forced the liquid storage chamber to release the drug through a single microneedle for transdermal drug delivery. The system could be finally loaded with a wireless drug release sensing device to realize the accurate and flexible release of drugs.

**Figure 2 pharmaceutics-14-00434-f002:**
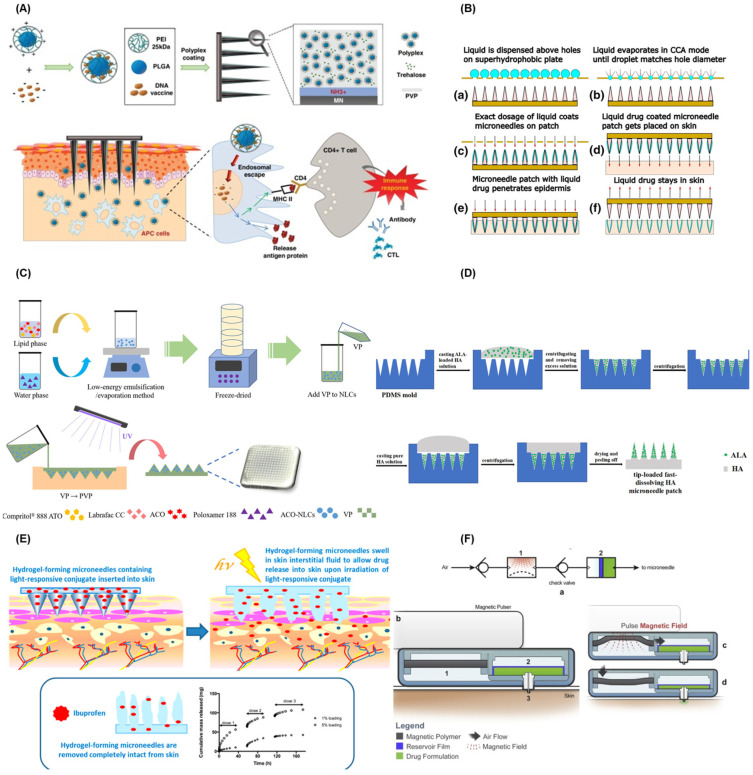
(**A**) Working principle diagram of ph1n1 DNA vaccine delivery platform of PLGA/PEI nanoparticle MNs. (**B**) Schematic diagram of MNs medical coating system. (a) liquid is dispensed above holes, (b) liquid evaporates in CCA mode until droplet matches hole diameter, (c) exact dosage of liquid coats MNs on patch, (d) liquid drug coated MN patch gets placed on skin, (e) MN patch with liquid drug penetrates epidermis and (f) liquid drug stays in skin. (**C**) Process diagram of the ACO–NLCs–MNs synthesis. (**D**) The process of fabricating ALA–HA fast-dissolving MNs. (**E**) Working principle of the light-responsive hydrogel MNs. (**F**) Process flow diagram of the magnetic polymer-driven MNs system. (**A**) HaeYong et al. [64]; (**B**) Andreas et al. [65]; (**C**) Guo et al. [67]; (**D**) Zhao et al. [68]; (**E**) John G. et al. [69]; (**F**) Jayaneththi et al. [74].

### 3.2. Micro-Reservoir System

The micro-reservoir system is composed of one or more drug reservoirs to realize drug storage. Compared with traditional drug release systems, the micro-reservoir system can realize a variety of delivery schemes such as zero-order, pulsatile, and on-demand dosing [42,75]. In particular, the micro-reservoir system manufactured by microfluidic technology can realize efficient and accurate drug delivery in vitro and in vivo [2,76]. Moreover, both in vitro and implanted micro-reservoirs can significantly increase drug stability and prolong the drug delivery time. The key part of the microfluidic micro-reservoir system is to prepare a driving system to meet the requirements of precise and stable release of drugs. Based on different actuation mechanisms, the micro-reservoir system can be actuated in active mode and passive mode [2,47,77].

#### 3.2.1. Active Actuation Mode for Micro-Reservoir System

In the active mode, a driving system is often required, which can be driven mechanically, magnetically, electrochemically, or by lasers. The biggest advantage of the active mode is that drug release can be accurately controlled, which can be easily carried out by patients for their own treatment [2,78,79]. Recently, more attention has been paid to the application of new technologies to drive drug delivery. Jeffrey Fong et al. [80] created an active, implantable wireless DDS which was driven by a radio-controlled pump for accurate drug delivery (Figure 3A). Their device was made of shape memory alloy [81] and driven by micro-electro-mechanical system (MEMS) technology. In this system, when the external radio-frequency [58] electromagnetic field was activated, the coil resonance would raise the temperature. Once the temperature exceeded the threshold of the alloy, the free-end alloy would return to flat from the curved shape, squeeze the pump chamber, and release the drugs from the drug release chamber. When the electric field was closed, with the recovery of the alloy, negative pressure was formed in the release chamber so that the drug could be supplemented from the storage chamber to the release chamber to achieve a cycle. The established platform provides the abilities for accurate drug release control and long-lasting drug delivery.

In some cases, some drugs need adjuvants or combination with other drugs to improve the final therapeutic effect, such as for tissue repair or regeneration and cancer therapy. For instance, Nobuhiro et al. [82] proved an electrically controlled dual-drug delivery polymeric platform, which is made of tri-(ethyleneglycol)-dimethacrylate (TEGDM) and poly (ethyleneglycol) dimethacrylate (PEGDM), as seen in Figure 3B. This device combined edaravone (EDV) and unoprostone isopropyl (UNO) to cure eye diseases, and the release of different compounds could be controlled by varying the TEGDM/PEGDM ratio. The results of in vivo and in vitro tests showed a good therapeutic effect on light-induced retinal damage in rats. With the popularity of mobile devices and the trends in interdisciplinary practices, micro-reservoir delivery systems based on smartphones have gained increasing interest and attention in recent years. Deshpande et al. [83] introduced an operable artificial pancreas (AP) app to release insulin for diabetic patient treatment and combined it with a smartphone to continuously monitor glucose in real time. It is the first AP app applied to human evaluation on an unlocked smartphone. The test results show that their system can realize safe and effective blood glucose adjustment.

#### 3.2.2. Passive Actuation Mode for Micro-Reservoir System

In the passive mode, the drugs are released by osmotic potential, diffusion transport, concentration gradient, and many other stimulations of environmental response [2,78,79]. A stable and prolonged drug release curve can be realized in the passive mode under preset conditions. Generally speaking, the manufacturing of passive-mode-driven micro-reservoir microfluidic DDSs is usually uncomplex and easy to manufacture, and there is no need for external power supply. For example, Yang et al. [84] designed a “reservoir-microfluidic channel” system based on the passive mode to realize multiple-drug delivery in a sequential manner. The drug release rate, sequence, and pH responsiveness can be precisely controlled by the length, width, and straightness of the micro-reservoirs and microchannels through passive diffusion behavior (Figure 3C). Besides, the passive mode is usually used when self-regulation is needed. Lee et al. [85] proposed a wet microcontact printing (μCP) system to store liquid drugs. As shown in Figure 3D, the drug solution was transferred to the micro-reservoir by the wet μCP. They explored the impact of many factors on drug delivery, such as the number of printings, the size of the printed picture, drug concentration, the materials of the carrier, and so on. The results of the drug-transferring experiments with their simple and accessible device showed good drug delivery performance.

**Figure 3 pharmaceutics-14-00434-f003:**
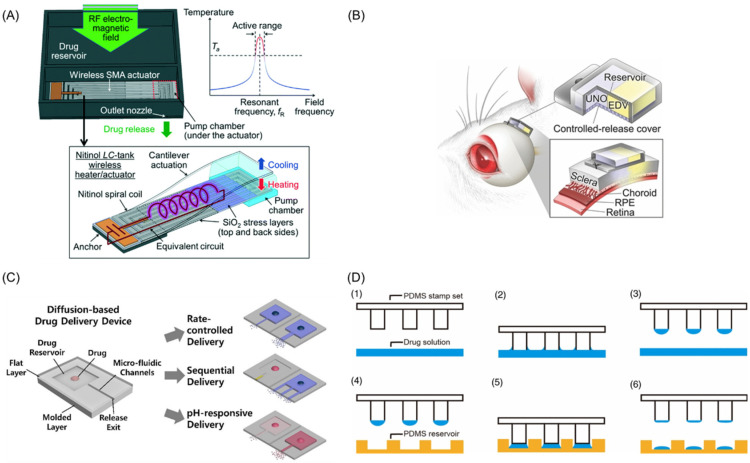
(**A**) Schematic diagram of wireless implantable magnetic driven DDSs and its working principle. (**B**) Working diagram of dual-drug delivery system. (**C**) The “reservoir-microfluidic channel” system optimization and drug release diagram. (**D**) Schematic diagram of the wet μCP system manufacturing: (1) PDMS stamp set and drug solution, (2) tip contact with drug solution, (3) drug coated stamps, (4) The position of drug coated stamp above PDMS reservoir, (5) stamp contact on a target surface and (6) final drug loading formation. (**A**) Jeffrey Fong et al. [80]; (**B**) Nobuhiro et al. [82]; (**C**) Yang et al. [84]; (**D**) Lee et al. [85].

## 4. In Vitro Models of Microfluidic Devices for the Evaluation of Drug Delivery

After the fabrication of nanomaterials for DDSs, the following critical work is to evaluate their drug delivery efficiency. In recent decades, a large number of DDSs have been constructed, but only a small amount succeed in translating from bench to clinic. One of the main reasons is that drug efficacy is largely affected by DDSs. When drugs enter the body, it is difficult to enter the blood circulatory system and transport to specific targets due to the influence of drug barriers and the first-pass effect. There are four types of biological barriers, namely the blood–brain barrier, the mucosal diffusion barrier, the biochemical barrier, and the cellular permeability barrier. The conventional methods are difficult to use to mimic the biological barriers for DDS evaluation. The commonly used methods are based on animal models but show poor predictive results for human responses to drugs. To overcome these problems, it is necessary to develop accurate and highly efficient strategies for the evaluation of DDSs. Recent developments in microfluidics, especially organs-on-a-chip, provide powerful tools for dynamic drug activity evaluation and whole-body responses in real time. An organ-on-a-chip is a platform in which different types of cells are co-cultured on a chip following the tissue-specific tridimensional to simulate a given single- or multi-organ system. It provides a high level of convenience for drug-delivery-related studies, which reduces the usage of animal experiments and fills the gap between animal studies and clinical trials to some extent [81]. In this section, we mainly focus on the mucosal model, vessel model, gut model, and blood–brain barrier models developed in recent years.

### 4.1. Mucosal Diffusion Barrier Model

Having the largest surface area (about 300–400 m^2^) in our body, the mucosal diffusion barrier is a significant obstruction for the uptake and absorption of drug carriers. Secreted by goblet cells, mucus plays a major role in promoting digestion, protecting the epithelium from harmful substances or bacteria, and enhancing the exchange of nutrients [86]. The mucin network can form a size exclusion filter for larger compounds (Figure 4B) [87]. Therefore, the mucosal diffusion barrier is considered an important effector that influences drug absorption and bioavailability. Several novel strategies have been applied to achieve effective mucosal drug delivery and sufficient drug bioavailability. Among them, using nanoparticles as drug carriers is widespread, being proven to protect drugs from degradation in the GI tract as well as the gastric environment [88,89]. As the diffusion capacity through mucus is highly dependent on hydrodynamic diameters and biointerfacial properties of nanoparticles, further understanding of the transportation mechanisms through mucus layers would highly accelerate the research and development process. Jia et al. [90] developed an in vitro mucus-model-on-a-chip by confining the mucin solution within the microchannel, forming a robust and reproducible model of mucus–aqueous solution interface, which can be used to readily observe the transport of nanoparticles under fluorescence microscopy. Result showed that the biointerfacial properties of mucus are critical for diffusion. Apart from spontaneous diffusion, biochemical and physical membrane disruption methods have been proposed. Ramesan et al. [91] explored the usage of high-frequency sound waves to promote drug diffusion through mucosa and controllably locate the drug in the mucosa of a porcine buccal model. By tuning the system parameters, the penetration depths can be changed so that over 95% of drugs are located within the mucosal layer with the preservation of their structural integrity. As the epithelial cells perform an important role in drug absorption, thorough simulation of human epithelia and mucosae models is extremely important [92]. From single epithelial layers to complex 3D models, cell-based mucus models provide an enhanced bio-relevant in vitro platform to better resemble physiological conditions. Hagiwara et al. [93] proposed a microphysiological-system-on-a-chip with 3D cultured Caco-2 tubules to mimic the gastrointestinal environment for the study of drug permeability in the stomach. Assessment using this model can reproduce similar results to those obtained from conventional methods. Gholizadeh et al. [94] developed a nasal-epithelial-mucosa-on-a-chip (NEM-on-a-chip) model integrated with a novel carbon-nanofibers-modified carbon electrode, aiming to monitor real-time quantitative drug transport rate in the nasal environment. Results indicated the great importance of using NEM-on-a-chip to emulate dynamic in vivo conditions. As discussed above, a model with the presence of epithelial cells and the right amount of mucus is needed for the development of a useful platform to test the nanocarrier.

### 4.2. Vessel Model

Blood vessels are one of the most important aspects of drug delivery, as most drugs are delivered by blood [95]. As an emerging platform for rapidly and accurately predicting in vivo behaviors of DDSs, the vessel-on-a-chip can provide insight to nanoparticles’ hemodynamics in microcirculation through blood. Generally, vessels-on-a-chip are based on the permeability and biocompatibility of the substrate materials, such as polydimethylsiloxane (PDMS) and hydrogel. Through the construction of a supporting structure in the microchannel, endothelial cells can be embedded to form vasculature. Mimicking the tumor microenvironment and vasculature system on a chip is beneficial for the direct and real-time study of drug delivery pathways. Studies on vessels-on-a-chip aim to understand the behavior of nanocarriers in vivo, especially the selectively extravasate and accumulative capacity. By using the co-culture of blood vessel cells in the microchannel to study the interaction between blood vessel cells and nanoparticles, the effects of vascular targeting, enhanced permeation, and retention [96] could be revealed [23]. Vascular targeting or particle targeting is a complex process in which intravascularly administered nanoparticles navigate the circulation and adhere to or cross the vessel wall to localize at their target site. Related research has attracted great attention to simulating fluid dynamics in the microvasculature and recapitulating the in vivo transportation process. Based on the establishment of the tumor microenvironment on a chip, fluorescently labeled gold nanoparticles were taken to trace and evaluate the whole transport processes. Several studies indicated that nanoparticle localization is highly dependent on vessel geometry and local flow features, especially shear stress. Generated by blood flow, shear stress has a great effect on transportation, accumulation, cellular uptake, and ultimately therapeutic efficacy [97]. For example, high-shear regions will adhere more strongly adhesive particles, and the weakly adhesive particles can be used for recirculation flows. Maria et al. [98] mapped the deposition of nanoparticles in a reconstructed vessel model. The result showed that the spatial distribution of the nanoparticles was dependent on physiological conditions and hemodynamic structures. The periodic changes of shear stress in vivo can prevent the adhesion of the nanoparticles. In addition to the vessel geometry, the size and shape of nanoparticles also have an impact on drug delivery efficiency. As the target site of drugs is always in the tissue beyond the wall, the function of margination and wall adhesion is pre-requisite, known as “near wall excess”. Nanoparticles that do not exhibit “near wall excess” in microvessels may be prevented from acting as drug delivery carriers. Cooley et al. [99] utilized four distinct shapes of nanoparticles, indicating that micro-scale non-spherical particles undergo enhanced margination and adhesion effect. In this study, a parallel plate flow chamber with red blood cells was established to observe the particle adhesion and retention in blood vessels. Considered to be the standard of drug delivery, the EPR effect was affected by vascular dynamics and nanoparticle design. Maneesha [23] developed an in vitro cancer-model-on-a-chip containing tumor cells and a vascular network to analyze the uptake of gold nanoclusters in tumor cells. Results indicated that cells near the endothelial gap absorbed more nanoparticles, which may be attributed to the leaky nature of tumor vasculature and the EPR effect. Moreover, the tumor size, number, and location would also affect drug transport and distribution. From Figure 5C, a tumor-vasculature-on-a-chip, which is constructed with a blood vessel channel and a tumor channel sandwiched with a porous membrane, was used to conduct numerical investigations [100] and revealed the influence of drug concentration heterogeneity in tumors.

Furthermore, 3D bioprinting technology can fabricate more complex architectures, which is a promising approach to improving the degree of biomimetic quality. For example, Cao et al. [101] proposed an improved hollow blood vessel along with a lymphatic vessel pair hosted in a 3D tumor microenvironment-mimetic hydrogel matrix. With 3D bioprinting technology, vessel permeability can be tuned individually, which is benefit for the diffusion profile study of biomolecules and anticancer drugs (Figure 5A). To improve drug delivery across the blood vessel, ultrasound is another feasible approach [102]. The cavitation effect caused by ultrasound is another approach to changing the permeability of vessels. A blood-vessel-on-a-chip [102] consisting of one tissue chamber and two independent vascular channels has been used to mimic tumor microvasculature and enable further studies on ultrasound-driven delivery, as it can temporarily open the endothelial intercellular junctions [96].

### 4.3. Gut-on-a-Chip

As the most important digestive organ, the gut plays a vital role in food digestion, absorption, and metabolism [103], provided by the villi and microvilli, and contains symbiotic microbial flora [104]. Generally, the intestinal barrier is comprised of mucus, the intestinal epithelium, and the biochemical environment [105], which represents the major barrier for drug absorption. For the intestinal epithelium barrier, there are four kinds of intestinal epithelial cells that are responsible for absorption (enterocyte), mucus secretion (goblet cells), hormone secretion (enteroendocrine cells), and defensive peptide secretion (Paneth cells), respectively [106]. The cell membranes of these epithelial cells form a physical barrier, which contributes to the selective transportation and efficiently protects our body from harmful substances or bacteria. In general, the membrane only allows hydrophilic solutes to permeate, but the proteins in the membrane can function as specific transporters, allowing nutrients to pass through. Besides, the tight junctions between adjacent epithelial cells play an important role in nutrient exchange as well as drug absorption. They are in charge of the integrity of the intestinal epithelium barrier and regulate the permeability of nutrients through two ways: the “pore” pathway, which is highly selective, and the “leak” pathway, which limits selectivity [107]. Besides, mucus secreted by goblet cells is considered an important effector that influences drug absorption and bioavailability. It is composed of hydrogel with mucin and a small number of proteins and carbohydrates. When drugs enter the mucus layer, mucus will be secreted and shed continuously to eliminate the attachment of drugs. As shown in Figure 4B [108], the mucin network can form a size exclusion filter for larger compounds [87].

Furthermore, the biochemical environment of the intestine, in which there exists a high concentration of digestive enzymes, also poses a great challenge for drug delivery. Therefore, therapeutic drugs, especially through oral drug delivery, generally face the problem of degradation, leading to unsustainable and un-targeted release. The development of biocompatible anti-digestive biomaterials provides promising potential to deal with this challenge. Ren’s research group synthesized an injectable covalent hydrogel through the photopolymerization of glycidyl methacrylate-modified xanthan (xan-GMA) on a microfluidic chip and cultured intestinal epithelial cells-6 on its surface to perform a gut barrier function [109]. The xan-GMA hydrogel showed good anti-digestion compared with an existing product, fibrin sealant, and showed good potential for closing gastrointestinal fistula. It is expected to be an outstanding gut repair material and drug delivery carrier. To further mimic the microenvironment and diversified function of intestine in vitro, in this study, a 2D gut microfluidic chip was used to replace the animal model of GI fistula. However, lacking 3D tissue architecture and cell–cell interactions makes it difficult to effectively recapitulate the real intestinal microenvironment and mimic the diversified function and structure of the gut with a 2D model. As a result, the 3D and dynamic culture models, made by co-culturing different kinds of cells in microchannels, are needed. Lee et al. [110] have proposed a 3D gut–liver chip to predict the first-pass metabolism. Two separate layer chips were used for the co-culture of gut (Caco-2) and liver (HepG2) cell lines in both 2D and 3D modes to reproduce a similar human PK profile and predict the pharmacological effects of drugs. This study demonstrated the importance of the employment of the multi-organ chip system. Herland et al. [111] proposed a multi-organ-chip system containing intestine, liver, and kidney cells to study drug pharmacokinetics (PKs) and pharmacodynamics (PDs) in vitro. The result obtained using this model is in accordance with the previously reported patient data, showing the accurate forecasting ability of pharmacodynamics.

### 4.4. Blood–Brain Barrier (BBB)

The BBB is a specific structure that protects our brain from harmful agents but results in bad drug delivery [112]. It functions as a selective physiological barrier which controls substance transport and maintains brain homeostasis [113]. The selective barrier works mainly through endothelial cells [114] in the tight junctions, the basement membrane of pericytes, and the end-feet of astrocytes [115]. From Figure 4A and Figure 5B, we can see that these cells are not functionally separated but are involved in complex activities and interactions among BBB-related cells because of the integrity of the BBB. For example, tightly connected ECs are able to prevent large molecules (approximately greater than 400 Da) from crossing the barrier by paracellular transport. Pericytes and astrocytes are closely attached to ECs and regulate the permeability of the tight junctions by releasing certain chemicals [116]. As a result, the BBB only allows a few water-soluble substances to enter the brain by active transport. However, some gas molecules and lipid-soluble substances can easily cross through passive diffusion [116]. Because of the existence of the BBB, many anticancer drugs fail to transport into the brain and cannot effectively treat brain tumors and nervous system diseases.

In order to overcome the poor BBB penetration of drugs, various approaches have been developed to enhance the transport efficiency and further improve the drug efficacy [117]. For small molecules (e.g., <500 Da), the methods of their delivery include local invasive (direct injection/infusion) delivery, induction of enhanced permeability, and application of global physiological targeting strategies [118]. For large hydrophilic molecules (e.g., enzymes, RNAi, genes), they may be degraded by the endosomal/lysosomal or ubiquitin/proteasomal system after internalization. In this case, some systems (e.g., endosomal escape mechanisms) could be applied to bypass these routes of degradation [119], and DDSs may help large molecules cross the BBB and enter the brain [120]. To date, the BBB still remains a bottleneck in brain drug development [121]. To boost the drug development of central nervous system diseases, a high-fidelity model of the BBB is necessary.

Compared to the conventional BBB in vitro models based on cell culture platform, the emerging BBB models on microfluidic devices have their own advantages of precise fluid control on a micro or nano scale and versatile integration capabilities. To evaluate different types of nanoparticles, various BBB models established by 2D and 3D culture modes on microfluidic devices have been reported. Wang et al. [121] proposed a pumpless microfluidic BBB model by the co-culture of brain microvascular endothelial cells and rat primary astrocytes on the two sides of a porous membrane (Figure 5D). The pumpless design was based on gravity-driven flow with minimized shear stress for long-term maintenance of BBB barrier function. The drug permeability of FITC-dextrans and model drugs was analyzed, with the results comparable to in vivo values. Eliana et al. [122] integrated the human primary astrocytes into a microfluidic platform and investigated the transmigration of T lymphocytes. To realize the parallel analytical performance, a bilayer chip with a 4 × 4 intersecting microchannel array was designed [123]. By the co-culture of neural endothelial cells and astrocytes in 16 BBB sites, this multisite BBB chip provided a more accurate prediction for drug permeability and toxicity through the BBB. Besides, 3D BBB models have been proposed to study brain-specific penetration’s mechanism using hydrogel as an extracellular matrix scaffold. Le et al. [124] developed a model to verify peptides’ receptor-mediated transcytosis and its cytotoxicity or cellular damage. Related quantification assays can be further performed for the evaluation of other brain-targeting drugs and carrier candidates. Related studies [125] also showed that a 3D BBB model including human induced pluripotent stem cell-derived endothelial cells, brain pericytes, and astrocytes exhibited more similar perusable and selective microvasculature than a rat brain. Such BBB models are more robust and physiologically relevant, which provides a more innovative and valuable platform for further drug discovery. Studies on the hypoxia-enhanced BBB chip further revealed the barrier’s function and studied the shuttling of drugs and antibodies [126]. Inducing pluripotent stem cells under hypoxic conditions can form the in vitro BBB-on-a-chip, and some relevant physiologically functions are exhibited for at least 1 week. The robustness of BBB-on-a-chip enables its application for testing nanoparticle transport mechanisms. Edwin et al. [127] have proposed a strategy by using fluorescence spectroscopy to analyze the transendothelial delivery of nanoparticles using a filter-free BBB model. Data confirmed that such a model is suitable for quantitatively studying nanoparticle transcytosis. Besides, Ahn et al. [128] fulfilled 3D mapping of nanoparticle distributions at cellular levels in their BBB model, demonstrating the distinction between cellular uptakes and BBB penetrations.

**Figure 4 pharmaceutics-14-00434-f004:**
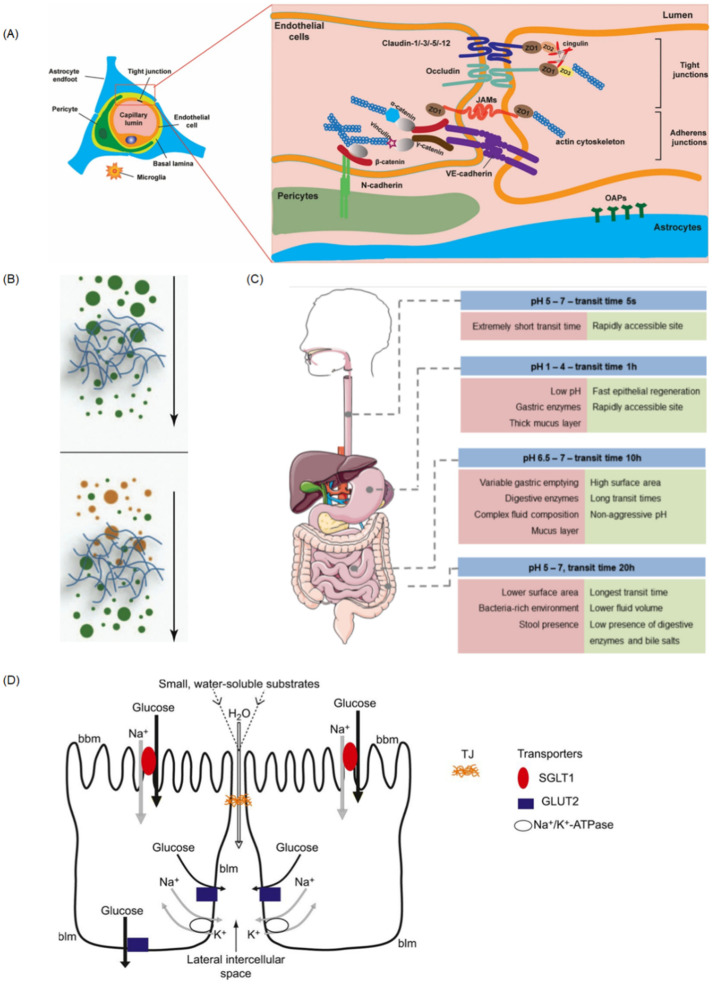
(**A**) Cells association at and molecular organization of the neurovascular unit (NVU). (**B**) The steric (**up**) and interactive (**down**) barrier properties of mucus. The mucus forms a filter, which can prevent diffusion across the gel according to size (**up**) or surface properties (**down**) of the diffusing compound. (**C**) Schematic representation of GIT sections and their pH, transit time, and relevant parameters for drug delivery. (**D**) Schematic diagram illustrating glucose absorption by transcellular pathways and paracellular pathways. Reprinted and adapted with permission from: (**A**) Kadry et al. [89], (**B**) Lieleg et al. [108], (**C**) Alonso et al. [129], and (**D**) Karasov et al. [130].

**Figure 5 pharmaceutics-14-00434-f005:**
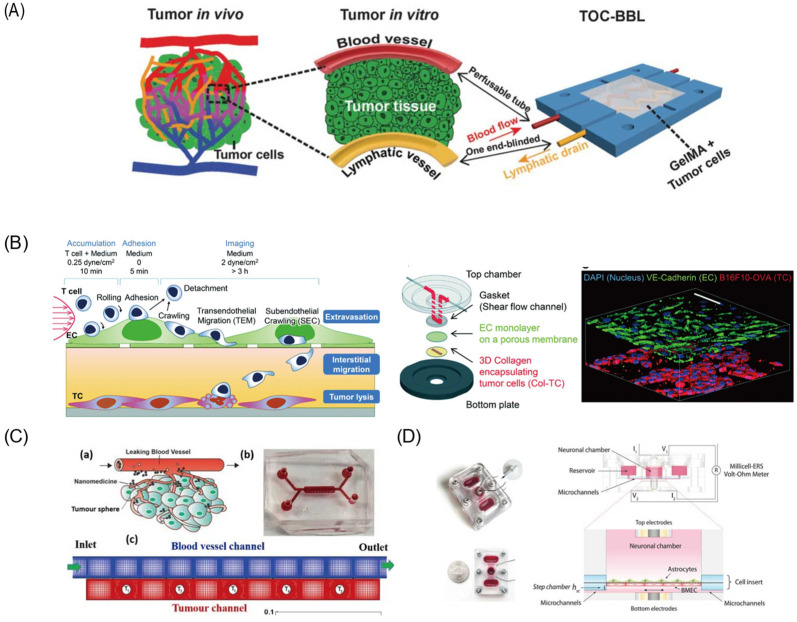
Different kinds of organs-on-a-chip. (**A**) The biomimetic in vitro model of tumor-on-a-chip with bio-printed blood and lymphatic vessel pair. (**B**) Multilayered blood vessel/tumor tissue chip (MBTC). (**C**) The schematic for the system of tumor tissue and blood vessels (a) schematic (b) picture of microfluidic device. (c) Computational mesh (**D**) The assembled microfluidic human model of BBB and its side view showing the fluid pathway. Reprinted and adapted with permission from: (**A**) Cao et al. [101], (**B**) Lee et al. [114], (**C**) Li et al. [100], and (**D**) Wang et al. [121].

## 5. Conclusions and Prospective

Having witnessed the rapid advances in the establishment of DDSs by microfluidics, the bioavailability, biocompatibility, and therapeutic index of drugs have undergone a huge improvement. Fabricated by microfluidics technologies, nanoparticles show superior performance in size distribution, drug encapsulation efficiency, and circulation time. The production of nanoparticles is reproducible, as the parameters can be precisely regulated in microfluidics. However, there are still several critical challenges in translating them from academic research to industrial and clinical practice. The most important issue is the capacity of the microfluidic platform. Parallelization of the design would be critical for larger-scale application. Avoiding complex processes in the fabrication of drug carriers, a carrier-free system may be more likely to achieve practical application. Particularly, many microdevices we mentioned above are implantable and have gone through in vivo testing. However, further advanced integration into microfluidic platforms is still required to fulfill diverse functions, such as self-tuning dynamic drug delivering and personalized drug delivery for the microneedle systems. It is believed that the carrier-free microfluidic DDSs can have a broader development in the future with the help of interdisciplinary collaboration.

Besides, the organ chip has become a more and more powerful platform for screening and evaluating DDSs, as it enables direct control of the biomechanical, biochemical, and biophysiological microenvironment, which shows great potential to reduce the reliance on animal models. The convergence of this technology with nanomedicine would offer a promising approach for more accurate and reliable preclinical assessment of new DDSs for therapeutic applications.

## Data Availability

This study did not report any data.

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
