# Peer review of "Recent Development of Drug Delivery Systems through Microfluidics: From Synthesis to Evaluation"

_pharmaceutics, 2022, doi:10.3390/pharmaceutics14020434_

Round 1

Reviewer 1 Report

The manuscript designed, covered and explained the application of microfluidics in the production of formulations. I recommend this interesting and well organized work for publication. However, the following few points need to be addressed.

In title, how the microfluidics applied in the synthesis of the drug delivery systems not cleared in the text. The technology mainly focused on the drug product preparation. Rewrite the titel.

I suggest the authors, write the large scale production applications of microfluidics in the commercial industry setup.

Include a brief note on recent patents on application of microfluidics technology based products.

Comment 1: line 10: “rapidly excretion” needs correction.

Comment 2: line 10-11 “rapidly excretion when crossing many biological barriers, resulting in a small amount of drugs at the pathological sites”. This sentence needs clarification.

Comment 3: line 41-42. “Known as 1st generation drug delivery system, hypodermic injections, oral administration, and inhalation methods possess several drawbacks on targeting delivery and the risk of degradation”. Authors are mixing here between drug delivery system and route of administration, and this is also not an absolute fact because many investigations have easily targeted drugs without microfluidics, therefore the authors should compare to conventional dosage forms not to kill them.

Comment 4: the authors should clearly define microfluidic platforms in the introduction section.

Comment: line 628 “various of approaches” needs correction.

Author Response

Reviewer #1:

The manuscript designed, covered and explained the application of microfluidics in the production of formulations. I recommend this interesting and well-organized work for publication. However, the following few points need to be addressed.

 Q1. In title, how the microfluidics applied in the synthesis of the drug delivery systems not cleared in the text. The technology mainly focused on the drug product preparation. Rewrite the title.

Answer: Thanks a lot for your kindly suggestion. We agree the reviewer’s suggestion, the title of our manuscript is not very good, we have changed the title as “Recent development of drug delivery systems on microfluidics: from synthesis to evaluation”. As for the way microfluidics applied in the synthesis of the drug delivery systems, we explain it in line 109-118.

  Q2. I suggest the authors, write the large-scale production applications of microfluidics in the commercial industry setup. Include a brief note on recent patents on application of microfluidics technology-based products.

Answer: Thank you very much for your careful review and kindly suggestion. We have already added this part in our manuscript line 49-65. 

Q3. rapidly excretion when crossing many biological barriers, resulting in a small amount of drugs at the pathological sites”. This sentence needs clarification.

Answer: Thank you for your kindly suggestions. We have reinterpreted this sentence by clarified the mechanism of the prevention of biological barriers. It was rewritten “Conventional drug administrations usually face the problems of degradation and rapid excretion when crossing many biological barriers, leading only a small amount of drugs arrive at the pathological sites.” 

Q4. Known as 1st generation drug delivery system, hypodermic injections, oral administration, and inhalation methods possess several drawbacks on targeting delivery and the risk of degradation”. Authors are mixing here between drug delivery system and route of administration, and this is also not an absolute fact because many investigations have easily targeted drugs without microfluidics, therefore the authors should compare to conventional dosage forms not to kill them.

Answer: We are sorry for our ambiguous description. We have corrected this mistake. Basically, drug delivery systems are aiming for the technology that helping targeted delivery and controlled release of therapeutic agents. The DDSs need to adapt different route of administration as the main barriers would change along with them. We rewritten this part in line 44-49 by clearly distinguish the relationship between conventional drug delivery methods and route of administration, emphasizing the shortcoming of conventional drug delivery methods.

Q5. the authors should clearly define microfluidic platforms in the introduction section.

Answer: Thank you very much for the good suggestion, we have added this part in the introduction section in line 49-55.

Reviewer 2 Report

This review aims to present the recent advances in the synthesis and evaluation of drug delivery systems on microfluidics, tackling various subjects, such as nano-drug delivery systems (liposomes, polymers, and inorganic compounds), micro-reservoir and microneedles drug delivery systems, some in vitro models on the micro-fluidic devices (blood-brain barrier model, vascular model, small intestine model). The proposed subject is of great interest, such microfluidic platforms having prospective applications in various clinical applications.

However, a major concern is represented by the degree of novelty brought by this work, and its ability to impact the reader, since there are numerous recent review articles presenting mostly the same information. Even the titles are quite similar, for example, “Recent Advances in Microfluidics for the Preparation of Drug and Gene Delivery Systems” Mol. Pharmaceutics 2020, 17, 12, 4421–4434; “Recent Advances of Controlled Drug Delivery Using Microfluidic Platforms” Adv Drug Deliv Rev. 2018 Mar 15; 128: 3–28; “Recent Advances of Microfluidic Platforms for Controlled Drug Delivery in Nanomedicine” Drug Des Devel Ther. 2021; 15: 3881–3891; “Recent developments in microfluidic technology for synthesis and toxicity-efficiency studies of biomedical nanomaterials” Materials Today Advances, vol. 13, 2022, 100205.

Additionally, extensive editing of English language and style is required, taking into account (not only) the minor comments listed below:

L10 Rapidly – rapid

L35 various of drug carriers

L62 Directly - direct

L65 system – systems; the drugs are released

L67 leading – lead

L90 includes

L93 itself - themselves

Author Response

Reviewer #2:

This review aims to present the recent advances in the synthesis and evaluation of drug delivery systems on microfluidics, tackling various subjects, such as nano-drug delivery systems (liposomes, polymers, and inorganic compounds), micro-reservoir and microneedles drug delivery systems, some in vitro models on the micro-fluidic devices (blood-brain barrier model, vascular model, small intestine model). The proposed subject is of great interest, such microfluidic platforms having prospective applications in various clinical applications. However, a major concern is represented by the degree of novelty brought by this work, and its ability to impact the reader, since there are numerous recent review articles presenting mostly the same information Even the titles are quite similar, for example, Recent Advances in Microfluidics for the Preparation of Drug and Gene Delivery Systems” Mol. Pharmaceutics 2020, 17, 12, 4421–4434; Recent Advances of Controlled Drug Delivery Using Microfluidic Platforms” Adv Drug Deliv Rev. 2018 Mar 15; 128: 3–28; “Recent Advances of Microfluidic Platforms for Controlled Drug Delivery in Nanomedicine” Drug Des Devel Ther. 2021; 15: 3881–3891; Recent developments in microfluidic technology for synthesis and toxicity-efficiency studies of biomedical nanomaterials” Materials Today Advances, vol. 13, 2022, 100205.  

Answer: Thank you very much for the good suggestion. As for your concerned, we have compared those review mentioned above and summarized their main content as follow:

This review mainly focused on the principles of controlled rapid mixing for producing DDSs and recent applications in pharmaceutical preparations aiming at the problems of batch-to-batch variation and production rate existed in conventional synthesis methods (“Recent Advances in Microfluidics for the Preparation of Drug and Gene Delivery Systems” Mol. Pharmaceutics 2020, 17, 12, 4421–4434; ) .

This review mainly introduced the recent advances in numerous microfluidic DDSs including  drug carrier-free micro-reservoirbased drug delivery systems, highly integrated carrier-free microfluidic lab-on-a-chip systems, drug carrier-integrated microfluidic systems, and microneedles (“Recent Advances of Controlled Drug Delivery Using Microfluidic Platforms” Adv Drug Deliv Rev. 2018 Mar 15; 128: 3–28; “).

This review mainly summarized the established DDSs on microfluidics, corresponding fabrication strategies, and the current advances of in vitro models on microfluidics for fast screening of DDSs (“Recent Advances of Microfluidic Platforms for Controlled Drug Delivery in Nanomedicine” Drug Des Devel Ther. 2021; 15: 3881–3891”).

This review mainly focused on the synthesis of different kinds of nanoparticles for drug delivery and the recent developed organ-on-chip for the evaluation of toxicity and efficiency of nanoparticles (“Recent developments in microfluidic technology for synthesis and toxicity-efficiency studies of biomedical nanomaterials” Materials Today Advances, vol. 13, 2022, 100205.).

Although the reviews that the reviewer pointed out seem similar to our manuscript from the title, each of the above reviews only focuses on one or two aspects, such as the synthesis technology of DDSs carrier based on microfluidic chip or in vitro models on chip for the evaluation of DDSs. Our manuscript mainly summarized the recent advances in the past five years regarding to the synthesis of nano DDSs and carrier free DDSs, and the recent developed in vitro models on chip recapitulating four primary biological barriers that affect the drug delivery and drug efficiency.

In addition, we agree the reviewer’s suggestion, the title of our manuscript is not very good, we have changed the title as “Recent development of drug delivery systems on microfluidics: from synthesis to evaluation”

Round 2

Reviewer 1 Report

The manuscript modified as per the suggested edits/comments.

Reviewer 2 Report

The new proposed title suits better the manuscript.